# The effects of social determinants on children's health outcomes in Bangladesh slums through an intersectionality lens: An application of multilevel analysis of individual heterogeneity and discriminatory accuracy (MAIHDA)

**Proloy Barua**[1]*, **Eliud Kibuchi**[2], **Bachera Aktar**[1,3], **Sabrina Fatema Chowdhury**[1], **Imran Hossain Mithu**[1], **Zahidul Quayyum**[1], **Noemia Teixeira de Siqueira Filha**[4], **Alastair H. Leyland**[2], **Sabina Faiz Rashid**[1], **Linsay Gray**[2]

1 BRAC James P Grant School of Public Health, BRAC University, Dhaka, Bangladesh, 2 MRC/CSO Social and Public Health Sciences Unit, University of Glasgow, Glasgow, United Kingdom, 3 Liverpool School of Tropical Medicine, Liverpool, United Kingdom, 4 Department of Health Sciences, University of York, York, United Kingdom

* proloy.barua@bracu.ac.bd

## Abstract

Empirical evidence suggests that the health outcomes of children living in slums are poorer than those living in non-slums and other urban areas. Improving health especially among children under five years old (U5y) living in slums, requires a better understanding of the social determinants of health (SDoH) that drive their health outcomes. Therefore, we aim to investigate how SDoH collectively affects health outcomes of U5y living in Bangladesh slums through an intersectionality lens. We used data from the most recent national Urban Health Survey (UHS) 2013 covering urban populations in Dhaka, Chittagong, Khulna, Rajshahi, Barisal, Sylhet, and Rangpur divisions. We applied multilevel analysis of individual heterogeneity and discriminatory accuracy (MAIHDA) to estimate the Discriminatory Accuracy (DA) of the intersectional effects estimates using Variance Partition Coefficient (VPC) and the Area Under the Receiver Operating Characteristic Curve (AUC-ROC). We also assessed the Proportional Change in Variance (PCV) to calculate intersectional effects. We considered three health outcomes: cough, fever, and acute respiratory infections (ARI) in U5y. We found a low DA for cough (VPC = 0.77%, AUC-ROC = 61.90%), fever (VPC = 0.87%, AUC-ROC = 61.89%) and ARI (VPC = 1.32%, AUC-ROC = 66.36%) of intersectional strata suggesting that SDoH considered do not collectively differentiate U5y with a health outcome from those with and without a health outcome. The PCV for cough (85.90%), fever (78.42%) and ARI (69.77%) indicates the existence of moderate intersectional effects. We also found that SDoH factors such as slum location, mother's employment, age of household head, and household's garbage disposal system are associated with U5y health outcomes. The variables used in this analysis have low ability to distinguish between those with and without health outcomes. However, the existence of moderate

**Data Availability Statement:** We used secondary deidentified dataset. The dataset named "Bangladesh Urban Health Survey (UHS) 2013" which was used in this study is publicly available at the Website (https://dataverse.unc.edu/dataset.xhtml?persistentId=doi:10.15139/S3/12274) upon responsible request.

**Funding:** The GCRF Accountability for Informal Urban Equity Hub ('ARISE') is a UKRI Collective Fund award with award reference ES/S00811X/1. EK, AHL and LG are also funded by the Medical Research Council (MC_UU_00022/2) and the Scottish Government Chief Scientist Office (SPHSU17). The funders had no role in the design and conduct of the study, nor the decision to prepare and submit the manuscript for publication.

**Competing interests:** The authors have declared that no competing interests exist.

intersectional effect estimates indicates that U5y in some social groups have worse health outcomes compared to others. Therefore, policymakers need to consider different social groups when designing intervention policies aimed to improve U5y health outcomes in Bangladesh slums.

## Introduction

### Background

In the Bangladeshi context, the most widely accepted definition of slums (also known as informal urban settlements) is "settlements with a minimum of 10 households or a small and untidy housing unit with a minimum of 25 members and predominantly very poor quality; very high population density and room crowding; very poor environmental services, especially water and sanitation; very low socio-economic status; and lack of security of tenure" [1, 2]. According to the Census of Slum Areas and Floating Population 2014, 22,27,754 people live in different slums in Bangladesh with almost 48% living in Dhaka city (the capital of Bangladesh) [3] and of which 10% are children under five years old (U5y) [4].

The common causes of child mortality in Bangladesh slums are fever and acute respiratory infections (ARI) [5–7]. In this study, ARI is defined as cough accompanied by short, rapid, or difficult breathing which is chest related [4]. The prevalence of ARI (33%) and fever (36%) are higher among children living in urban slums compared to their rural counterparts in Bangladesh [8, 9]. There are also regional variations in the prevalence of childhood illnesses such as cough, fever, and ARI. For example, fever (41.6%) and cough (21.5%) are also common among all slum dwellers in Dhaka [5, 10] while ARI among children is most prevalent in Rajshahi (10.2%), Khulna (8.3%) and Barisal (8%) which are administrative divisions of Bangladesh [6]. The ARI hospitalisation rate among children in Khulna is 46.2% and is much higher among males (65.8%) compared to females at 36.2% [7]. Children living in Chittagong (6.1%), Sylhet (6.2%), and Rangpur (6.6%) regions are at higher risk of developing ARI [6].

U5y mortality varies in different divisions of Bangladesh. For instance, the highest U5y mortality is found in Chittagong division (79 per 1000) followed by Barisal (71 per 1000), Rajshahi (71 per 1000), Dhaka (69 per 1000) and Khulna division (58 per 1000) [11]. In addition, U5y mortality rates are higher in the slums (80.7%) than in non-slums (31%) in Bangladesh [9]. The higher susceptibility of these health outcomes for U5y living in Bangladesh slums impede both their physical and cognitive development which leads to poor health both currently and in the future [12].

The poor health outcomes experienced by U5y living in Bangladesh slums are often shaped by social determinants of health (SDoH) [13–15]. SDoH are described as the conditions in which people live, grow, learn, work, play, and age [13, 16]. For instance, unhygienic conditions, overcrowding, lack of formal health facilities, the transmission of vaccine-preventable diseases, low immunization coverage coupled with low socioeconomic status of parents make U5y more vulnerable in slums [17, 18]. SDoH also form social hierarchies that in turn determine the distribution of power, prestige, and resources among social groups [19]. People living in slums are usually excluded from social structures of power, denying their rights to access public resources, which leads to health inequities [20, 21]. Therefore, to promote health equity among U5y living in slums requires an understanding of how collectively SDoH shape their health outcomes through an intersectionality lens.

The concept of intersectionality describes the phenomenon in which social determinants such as gender, race, ethnicity, sexual orientation, religion, disability, class and other forms of identity "intersect" to create unique dynamics and effects [22]. When these identities do not

work independently but interact, they lead to aggravated social inequities. We have used the intersectionality concept to identify group level effects (termed as "intersectional effects" in this paper) and individual levels effects (termed as "main effects" in this paper) in under-five health outcomes [23].

Previous studies have investigated the associations between SDoH and health outcomes (i.e. diarrhoea, fever, cough and ARI) among U5y in Bangladesh [7, 8, 14, 15, 24–29]. They have found that child's age [7, 8, 14, 28, 29], child's gender [7, 14, 15], mother's employment [28], father's occupation [24], housing conditions [28], and low birth weight [29] are associated with ARI among U5y in the country. Similarly, low socio-economic status measured by the household wealth index [8, 14, 15, 24, 25, 27, 28], parental education [8, 14, 24, 27], rural residency and regional differences [8] are associated with fever, ARI, diarrhoea, and cough in U5y. In addition, the mother's nutritional status [24], mother's age [8, 14, 29], and child malnutrition [14, 24] are associated with diarrhoea and ARI in U5y. Finally, maternal depression [15], the household's access to improved water and sanitation [15, 27, 28], and the household's garbage disposal system [10, 15] are associated with ARI and cough in U5y. We aim to extend these analyses by evaluating the association of collective SDoH and health outcomes (in this case with or without fever, cough or ARI) for U5y via intersectional approach [23, 30].

## Objective

The primary analytic objective of this study is to explore how collectively SDoH affect health outcomes (i.e., fever, cough, and ARI) for U5y in Bangladesh slums.

## Methods

### Ethics statement

We used a secondary deidentified data set. Ethical approval was not required for the study. The dataset named "Bangladesh Urban Health Survey 2013 (UHS 2013)" which was used in this study is publicly available at the Website (https://dataverse.unc.edu/dataset.xhtml?persistentId=doi:10.15139/S3/12274) upon responsible request. For UHS 2013, ethical approval was obtained from the Bangladesh Medical Research Council and the Institutional Review Board at the School of Public Health, University of North Carolina at Chapel Hill [4].

### Data sources, participants, and settings

This study used UHS 2013 data, a representative cross-sectional household survey covering 53,700 households [4]. This is the most recent national survey covering urban population in Bangladesh and was implemented through a collaborative effort of four organisations [4]. The UHS 2013 data were collected between July 2013 and January 2014 using face-to-face interviews. The eligible respondents were all ever-married women aged 15–49 and ever-married men aged 15–54. Questions on child health were contained in the "General Child Health and Nutrition" section of the women's long questionnaire and administered to the women respondents [4]. We retrieved 3,173 records for U5y for our analysis after dropping 10 (0.31%) records due to missing values in cough. Similarly, we retrieved 3,183 records for fever and ARI in U5y without any missing data. Further details can be found in Kibuchi et al. [31].

The study area comprised of seven divisions: Barisal, Chittagong, Dhaka, Khulna, Rajshahi, Rangpur and Sylhet as shown in Fig 1 below. We combined Barisal, Chittagong, Rangpur or Sylhet divisions into one and named "Other divisions" because these are relatively small and new divisions except Chittagong. In addition, we needed to reduce the number of categories for simplifying analysis.

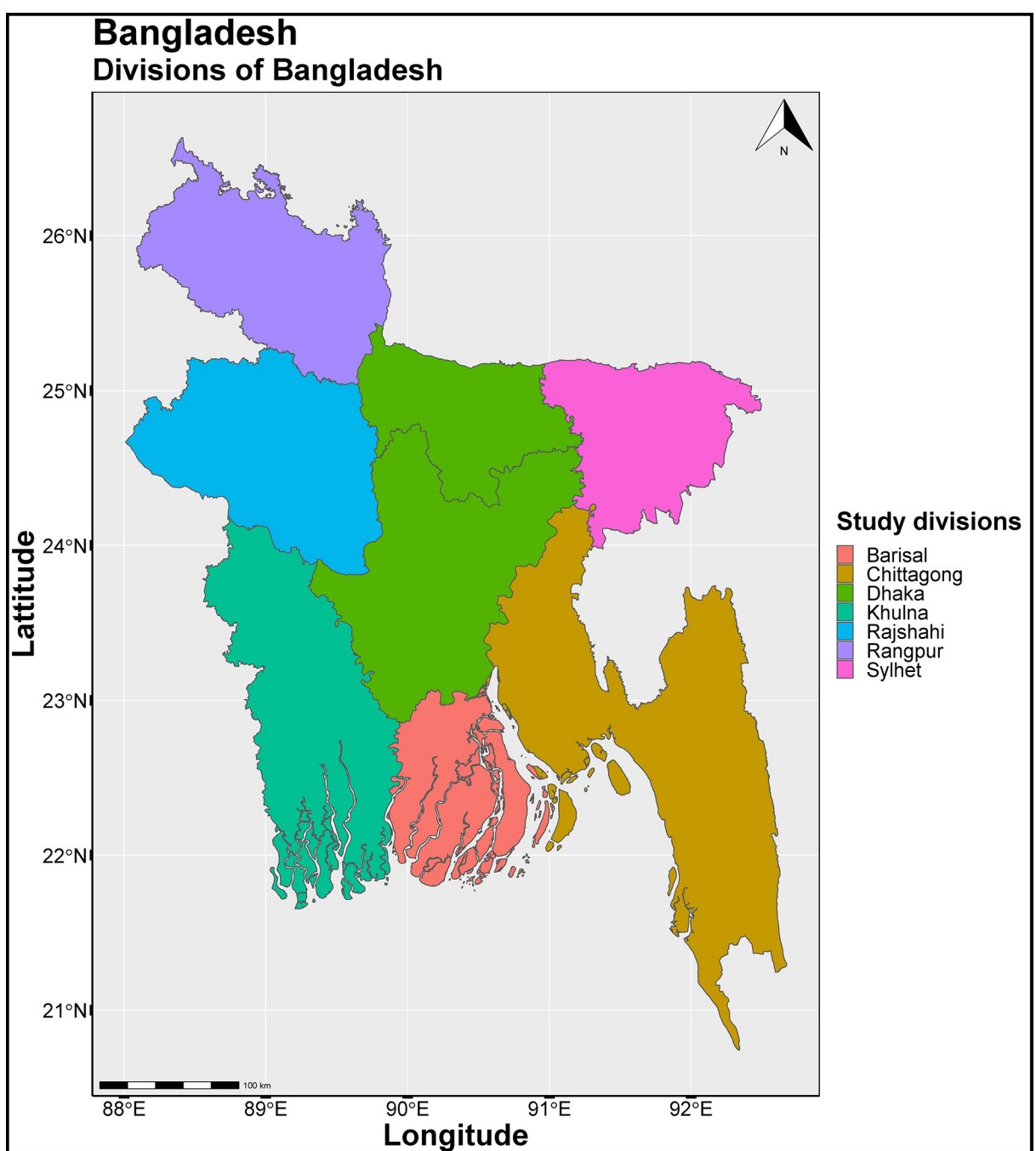

**Fig 1. Study area of Bangladesh Urban Health Survey 2013 (UHS 2013). UHS 2013 has been conducted in seven divisions of Bangladesh:** Barisal, Chittagong, Dhaka, Khulna, Rajshahi, Rangpur and Sylhet. R software was used to create the map. The source of the basemap shapefile is available on GADM website at: https://geodata.ucdavis.edu/gadm/gadm4.1/shp/gadm41_BGD_shp.zip. The data are freely available for academic use and other non-commercial use (CC-BY 2.0). The GADM license is available at https://gadm.org/license.html.

### Measures

The analytical approach used in this research requires all outcomes and explanatory variables to be categorical; hence continuous variables were grouped into categories [23]. The definitions and categorisations of outcome and predictor variables are provided in the S1 Table

The three health outcomes we considered for this study were: cough (whether a child had a cough or not), fever (whether a child had fever or not), and ARI. In UHS 2013, ARI is defined as cough accompanied by short, rapid, or difficult breathing, which is chest related [4]. The reference or recall period of these three health outcomes among U5y was the last two weeks from the day of the interview. Separate models were fitted for each of the three outcomes.

The explanatory variables used in the study were selected based on existing literature on factors that influence health outcomes for U5y as well as informed by the framework developed by the Commission for Social Determinants of Health (CSDH) [13, 32]. The explanatory variables were categorised into four groups: U5y demographic characteristics (e.g., age, sex), mothers' sociodemographic characteristics (e.g., age, ever attended school, employment, and marital status), head of households' sociodemographic characteristics (e.g., age, sex, marital status), and households' social structural characteristics (e.g., wealth index, housing condition, having separate kitchen, type of cooking fuel used, migration status, garbage disposal method, ownership of dwelling, and ownership of land) (S1 Table). We have used the wealth index variable already developed and available in the data set [33]. Further details on the choice of variables can be found in Kibuchi et al. [31]. The wealth index quintiles were constructed using Principal Components Analysis (PCA) [34]. The analysis was based on complete case analyses. We have selected social determinants of health (SDoH) based on their statistical significance with respective outcomes in the univariate analyses.

## Statistical methods

We applied multilevel analysis of individual heterogeneity and discriminatory accuracy (MAIHDA) [23, 35–38] to explore the effects of SDoH on three health outcomes via intersectionality. Unlike common measures of average association, MAIHDA allows for a better understanding of the heterogeneity of individual effects and inter-individual heterogeneity around group averages [23]. This method involves defining a number of intersectional strata based on combinations of social attributes such as a set of SDoH associated with their intersectional social identities and individual characteristics [31, 39].

MAIHDA approach has advantages over conventional fixed-effect approaches such as precision-weighted estimates, model parsimony, ease of interpretation, and reliable estimates for strata with small sample sizes [23, 35]. Additionally, MAIHDA allows for the inductive analysis of numerous stratum-specific interactions and it avoids pre-specifying specific interactions or intersections to explore [23, 35, 40, 41]. Most importantly, MAIHDA estimates the measures of Discriminatory Accuracy (DA) together with traditional comparison of population averages, enabling the most comprehensively informed decisions about suitable public health action to address an issue under study. MAIHDA provides a feasible way of measuring multiple intersections and analysing groups of small size from an intersectionality perspective. MAIDHA also provides a parsimonious way of analysing high dimensional interactions which is not possible with single level models [23, 38].

MAIHDA estimates DA and intersectional effect in two steps through two successive multilevel logistic regression models (Model 1 and Model 2). In the first step, Model 1 includes only random intercept without any predictor variables to estimate the mean of a health outcome and intersectional strata differences (i.e., variance). The Model 1 allows us to estimate the extent to which the variance in the health outcomes is explained by differences between intersectional strata versus differences within stratum via the Variance Partition Coefficient (VPC). The VPC presents as the percentage share of individual variance which lies within intersectional strata [42] and values higher than 5% indicating an acceptable DA [43]. A low VPC suggests that intersections have similar mean levels of a particular outcome and that individuals

differ substantially within intersectional strata, and vice-versa for a high VPC [23, 39]. The VPC of Model 1 represents the ceiling of the explanatory power of the intersectional strata and encompasses both additive and potential interactive effects of the variables that define the strata [43].

In the second step, in Model 2, the main effects (predictor variables used to construct intersectional strata) are added to the fixed part of the Model 1. The VPC now represents the extent to which intersectional clustering is multiplicative (e.g., aggravating effects). For instance, if the VPC reduces to around 0%, intersectional differences across strata are fully explained by the main effects suggesting additive effects rather than multiplicative effects. However, if the VPC in Model 2 is not equal to zero, it suggests the presence of interactional effects [39]. The main effects are presented as odds ratio (OR) to describe the association between the SDoH factors and health outcomes in U5y.

We also calculated DA using the Area Under the Receiver Operating Characteristic Curve (AUC-ROC) using the predicted probabilities obtained from each model [44, 45]. The AUC represents an overall accuracy of the model to classify children with or without a health outcome. The AUC-ROC values extend from 0.5 to 1, where 0.5 represents absence DA, and 1 represents perfect DA. The higher the AUC-ROC values above 0.5, the higher the influence of the intersectional strata to differentiate U5y with and without a health outcome (in this case, cough, fever and ARI) [46].

We also calculated the Proportional Change in Variance (PCV) to measure the overall proportion of intersectional strata variance of Model 1 that was explained after adding the main effects in Model 2 [43–45]. In the absence of any intersectional effects, the main effects would explain all strata variances (PCV = 100%) [36, 45, 47].

We also plotted intersectional effect estimates (stratum-level residual), and their corresponding 95% credible intervals (CIs) ranked from lowest to highest for both Model 1 and Model 2. A positive estimate suggests that children in an intersectional stratum have a higher risk than expected (e.g., disadvantaged group), while a negative intersectional estimate means a lower risk than expected (e.g., privileged or advantaged group) based on the addition of the risks factors associated with the variables that define the intersectional stratum [35, 48]. A residual of zero indicates that a social stratum represents exactly the mean outcome predicted by the additive main effects [48]. Further details on the application of MAIHDA for these analyses are described in the protocol document [31]. The statistical details can be found in the S1 File.

We used R statistical software (version 4.1.1) to fit MAIHDA models using the "brms" package (version 2.14.4) which is fully Bayesian inference through Markov Chain Monte Carlo (MCMC) methods [49]. We used weakly informative priors and ran all analyses using 50,000 iterations with a burning period of 5,000 [43]. MCMC chains were checked graphically for convergence [50].

## Results

### Sociodemographic characteristics of the population

We retrieved 3,183 children under five years old (U5y) from the UHS 2013 data set. Around 49% of U5y were male, and 51% were female. Around 37% of U5y were 0–1 years old, and 63% were 2–5 years old. Around 19%, 28%, and 5% of U5y were found with cough, fever, and ARI, respectively. The detailed socio demographic profile of the population can be found in the S2–S4 Tables. The following sections describe the results of three health outcomes in U5y such as cough, fever and ARI.

### Intersectional strata

We have created 147 intersectional strata for cough based on the following social positions of U5y (e.g., child age, division, age of household head, garbage disposal, and mother's employment), which were statistically significant in the univariate analyses (S5 Table). For fever, we have created 441 strata using child age, division, age of household head, mother's age, garbage disposal, mother ever attended school, mother's employment, and separate kitchen variables which were statistically significant in the univariate analyses (S6 Table). ARI had 247 strata based on child age, division, wealth index, age of household head, types of cooking fuel used, and mother's employment variables (S7 Table). The calculations of intersectional strata for each of the three outcomes can be found in S8 Table.

Table 1 presents models 1 and 2 based on the number of intersectional strata described above for each health outcome.

**Cough.**    In Model 1, the VPC is 5.3% indicating an acceptable DA. Only 5.3% of the total individual variance among U5y with and without cough are located at the intersectional strata level (Table 1). The AUC-ROC is 64.8%, suggesting a poor DA (Table 1). It indicates that intersectional strata have weak explanatory power to discriminate U5y with a cough from U5y and without a cough.

The VPC and the AUC-ROC have dropped to 0.8% and 61.9% respectively, after adjusting for main effects variables (e.g., child age, division, age of household head, garbage disposal, and mothers' employment) in Model 2. This indicates the existence of moderate intersectional effects estimates (Table 1). The PCV is 85.9% indicating that 14.1% of the total variance was not explained by adding main effects, which can be attributable to intersectional effects. This suggests that, the moderate intersectional inequalities in U5y of either having a cough or not are due to their social positions based on variables used to create intersectional strata.

The variables, child age, division, age of the household head, household garbage disposal method, and mother's employment had significant association with cough in U5y (Table 1). Older children aged between 2 to 5 years had 24% lower odds of cough than their younger counterparts aged 1 year or less. Compared to U5y living in Dhaka slums, the odds of having a cough were 1.9 times and 1.7 times higher among U5y living in slums in Rajshahi and other divisions respectively. U5y living with relatively older household heads (aged 30 to 44) had 38% lower odds of having cough. Furthermore, U5y from households having a better garbage disposal system (e.g., garbage collected from home, garbage disposed in bins outside of the house, and garbage disposed in open spaces) had 45%, 46% and 49%about 50% lower odds of having cough, respectively. Finally, mothers' employment had 25% lower odds of having cough in U5y (Table 1).

Fig 2 presents estimates of intersectional effects (stratum-level residuals) for cough with their corresponding 95% credible intervals (CIs) ranked from lowest to highest. The Panel A shows that the estimates for Model 1 ranging from -1.12 to 1.47, with only 4 (2.7%) of 147 intersectional strata's 95% CI not crossing zero (on the far left and far right), indicating significant intersectional effects estimates. Similarly, the Panel B represents intersectional effects estimates for Model 2 ranging from -0.51 to 0.63, and all intersectional strata's 95% CI having crossed zero. This indicates that although intersectional effects were present for cough among U5y, even after controlling for main effects variables, we did not find any significant intersectional effects for cough (Fig 2).

**Fever.**    In Model 1, the VPC is 3.9% indicating a low DA. This indicates, only 3.9% of the total individual variance among U5y with and without a fever are attributed to the intersectional strata level (Table 1). The AUC-ROC is 68.0%, suggesting a poor DA or intersectional

**Table 1. Main effects coefficient, strata variance, area under the curve, variance partition coefficient (VPC) and proportional change in variance (PCV) for cough (Model 1 and 2), fever (Model 1 and 2), and ARI (Model 1 and 2).**

| Variable | Category (Ref) | Cough | | | | Fever | | | | ARI | | | |
|---|---|---|---|---|---|---|---|---|---|---|---|---|---|
| | | Model 1 | | Model 2 | | Model 1 | | Model 2 | | Model 1 | | Model 2 | |
| | | Odds Ratio | 95% CI | Odds Ratio | 95% CI | Odds Ratio | 95% CI | Odds Ratio | 95% CI | Odds Ratio | 95% CI | Odds Ratio | 95% CI |
| Intercept | | 0.24* | (0.21,0.28) | 0.47* | (0.31, 0.71) | 0.39* | (0.36, 0.43) | 0.75 | (0.42,1.31) | 0.05* | (0.04,0.06) | 0.10* | (0.03, 0.27) |
| Child age | 1 year and less (Ref) | | | 1.00 | | | | 1.00 | | | | 1.00 | |
| | 2 to 5 | | | 0.76* | (0.64, 0.96) | | | 0.89 | (0.74,1.07) | | | 0.78 | (0.54, 1.12) |
| Division | Dhaka (Ref) | | | 1.00 | | | | | | | | 1.00 | |
| | Khulna | | | 1.41 | (0.91, 2.15) | | | 1.10 | (0.74,1.60) | | | 2.46* | (1.14, 5.17) |
| | Rajshahi | | | 1.94* | (1.00, 3.48) | | | 1.52 | (0.84,2.69) | | | 1.66 | (0.45, 4.98) |
| | Other divisions | | | 1.76* | (1.34, 2.16) | | | 1.50* | (1.22,1.84) | | | 2.06* | (1.39, 3.07) |
| Age of the household head | 13 to 29 years (Ref) | | | 1.00 | | | | 1.00 | | | | 1.00 | |
| | 30 to 44 years old | | | 0.62* | (0.54, 0.87) | | | 0.66* | (0.54,0.80) | | | 0.48* | (0.31, 0.73) |
| | 45 years and above | | | 0.81 | (0.66, 1.18) | | | 0.81 | (0.63,1.05) | | | 0.88 | (0.52, 1.43) |
| Garbage Disposal | Disposed within premises (Ref) | | | 1.00 | | | | 1.00 | | | | | |
| | Collected from home | | | 0.55* | (0.39, 0.90) | | | 0.65* | (0.45,0.95) | | | | |
| | Disposed in bin outside | | | 0.54* | (0.36, 0.89) | | | 0.63* | (0.42,0.95) | | | | |
| | Disposed in open spaces | | | 0.51* | (0.38,0.84) | | | 0.74 | (0.52,1.06) | | | | |
| Mothers' employment | No (Ref) | | | 1.00 | | | | 1.00 | | | | 1.00 | |
| | Yes | | | 0.75* | (0.60, 0.98) | | | 1.15 | (0.94,1.42) | | | 0.67 | (0.42, 1.06) |
| Mothers' age | Below 18 years (Ref) | | | | | | | 1.00 | | | | | |
| | 18 years and above | | | | | | | 0.72 | (0.46,1.14) | | | | |
| Mother ever attended school | No (Ref) | | | | | | | 1.00 | | | | | |
| | Yes | | | | | | | 0.84 | (0.68,1.03) | | | | |
| Separate kitchen | No (Ref) | | | | | | | 1.00 | | | | | |
| | Yes | | | | | | | 1.15 | (0.96,1.37) | | | | |
| Cooking fuel used | Charcoal, dung cakes (Ref) | | | | | | | | | | | 1.00 | |
| | Kerosene or liquid gas | | | | | | | | | | | 0.82 | (0.19, 3.06) |
| | Natural gas | | | | | | | | | | | 0.67 | (0.30, 1.63) |
| | Wood fuel | | | | | | | | | | | 0.48 | (0.21, 1.19) |

(*Continued*)

**Table 1.** (Continued)

| Variable | Category (Ref) | Cough Model 1 Odds Ratio | Cough Model 1 95% CI | Cough Model 2 Odds Ratio | Cough Model 2 95% CI | Fever Model 1 Odds Ratio | Fever Model 1 95% CI | Fever Model 2 Odds Ratio | Fever Model 2 95% CI | ARI Model 1 Odds Ratio | ARI Model 1 95% CI | ARI Model 2 Odds Ratio | ARI Model 2 95% CI |
|---|---|---|---|---|---|---|---|---|---|---|---|---|---|
| Wealth index | Rich (Ref) | | | | | | | | | | | 1.00 | |
| | Middle | | | | | | | | | | | 0.74 | (0.39, 1.40) |
| | Poor | | | | | | | | | | | 0.93 | (0.54, 1.66) |
| Variance | | 0.18 | | 0.05 | | 0.13 | | 0.02 | | 0.14 | | 0.04 | |
| Strata | | 147 | | 147 | | 441 | | 441 | | 247 | | 247 | |
| Observations | | 3,173 | | 3,173 | | 3,183 | | 3,183 | | 3,183 | | 3,183 | |
| AUC-ROC | | 64.8% | | 61.90% | | 68% | | 61.89% | | 72.93% | | 66.36% | |
| VPC | | 5.25% | | 0.77% | | 3.91% | | 0.87% | | 4.24% | | 1.32% | |
| PCV | | | | 85.90% | | | | 78.42% | | | | 69.77% | |

Ref: Reference category; 95% CI: 95% Credible Interval; AUC-ROC: Area Under the Receiver Operating Characteristic Curve; VPC: Variance Partition Coefficient.;

PCV: Proportional Change in Variance

* Statistically Significant at 5% level of significance.

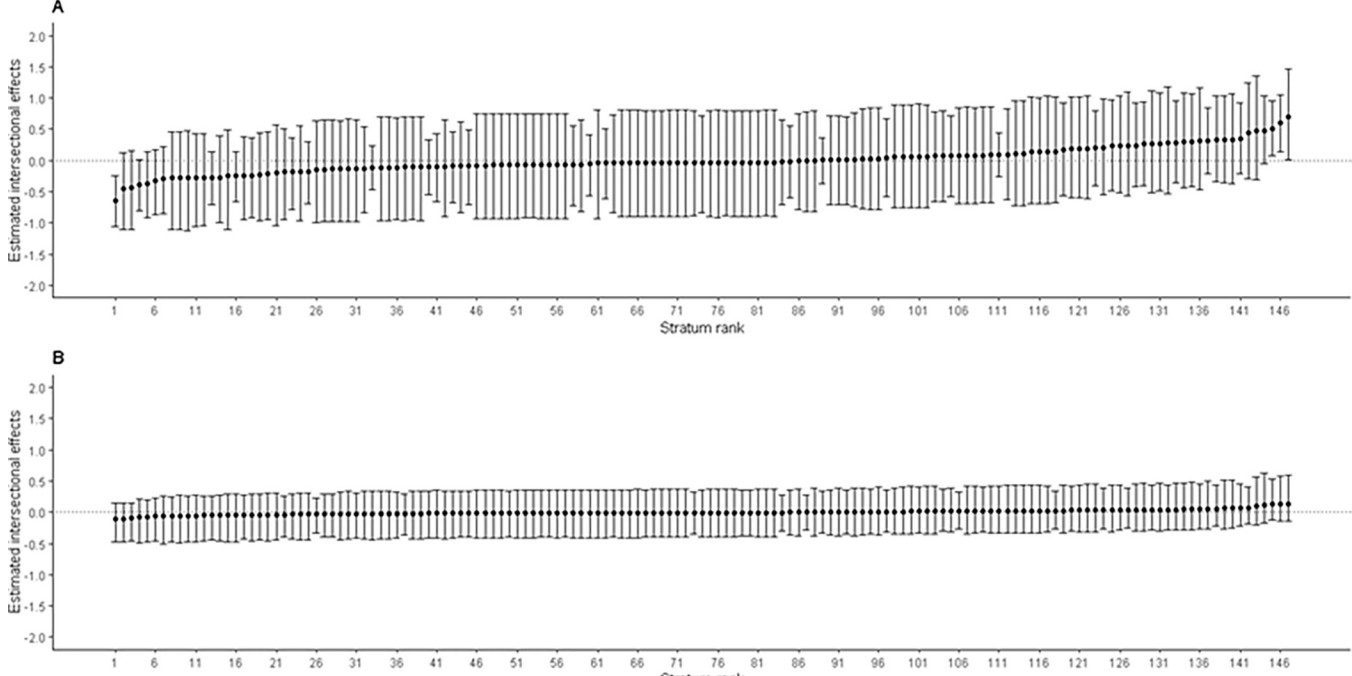

**Fig 2. Estimated intersectional effects (stratum-level residuals) and their corresponding 95% credible intervals (CIs) for each stratum ranked from lowest to highest for cough.** Panel A: An intercept model with only intersectional strata; Panel B: A Full model after adding main effects (variables used to create intersectional strata) with intercept model. Negative values indicate lower risk (privileged groups) of cough in under-five 'than expected' based on the additive contributions of the main effects, while positive values indicate higher risk (disadvantaged groups) of cough in under-five 'than expected' for that stratum.

strata have weak explanatory power to discriminate U5y with a fever from U5y without a fever.

The VPC and the AUC-ROC have dropped to 0.9% and 61.9%, respectively, after adjusting for main effects variables (e.g., child age, division, age of household head, mother's age, garbage disposal, mother ever attended school, mother's employment, and separate kitchen) in Model 2. The non-zero VPC indicates the existence of little multiplicative effects of intersectional strata in relation to the main effect variables included in the model. The AUC-ROC of 61.9% also indicates the low discriminatory power of intersectional strata to discriminate U5y with a fever from those without a fever. The PCV is 78.42% indicating that 21.6% of the total variance is not explained by adding main effects, which can be attributable to intersectional effects estimates (Table 1). The low DA suggests that most of the observed differences in fever across intersectional strata are due to the main effects, and only a small share of the differences is attributable to intersectional strata.

The main effects variables which were significantly associated with fever were division, age of household head, and household garbage disposal method are significantly associated with fever in U5y (Table 1). Children living in slums in other divisions were at a higher risk of getting a fever than those living in Dhaka slums. Children from households with relatively older household heads (aged 30 to 44) had 38% lower risk of having a fever. Finally, children from households with a better garbage disposal system (e.g., garbage collected from home, garbage disposed in bins outside of the house) had 35% and 37% lower risk of having a fever, respectively, compared to children from those households having a poor garbage disposal system (e.g., dispose garbage within the premises of the house) (Table 1).

Fig 3 presents estimates of intersectional effects (strum-level residuals) for fever with their corresponding 95% CI ranked from lowest to highest. The Panel A shows that the estimates

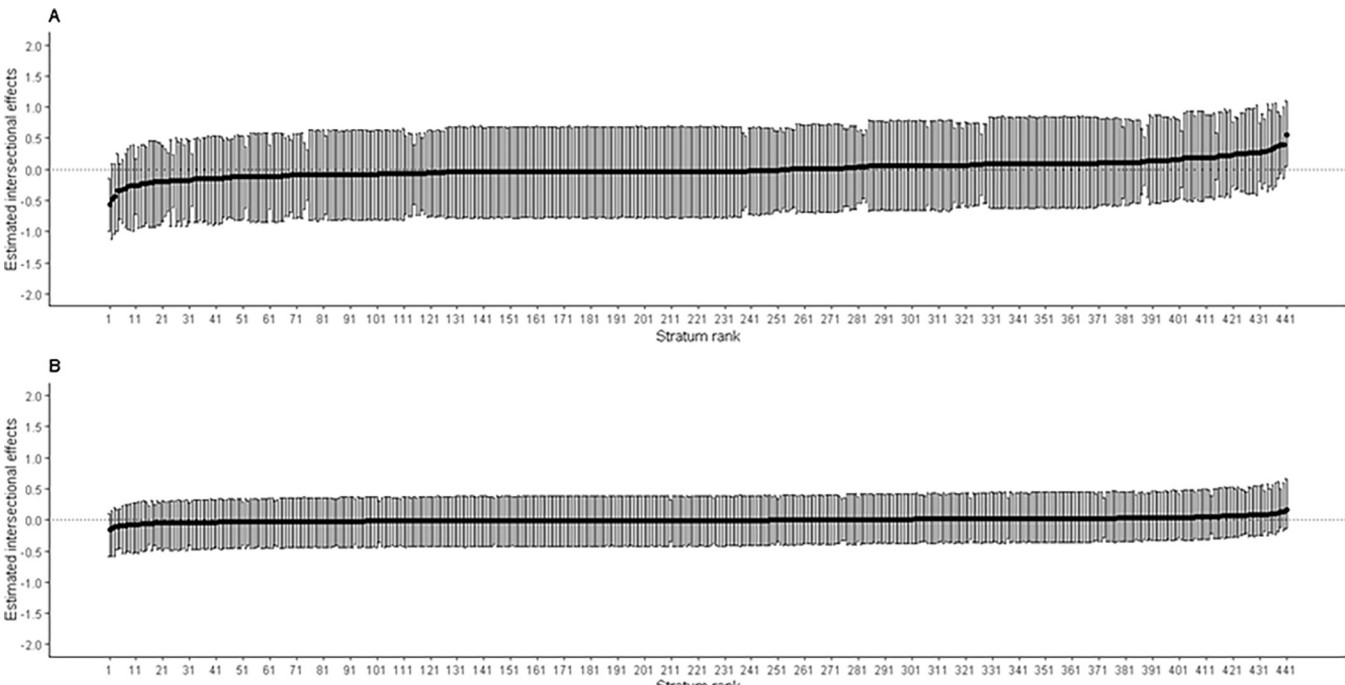

**Fig 3. Estimated intersectional effects (stratum-level residuals) and their corresponding 95% credible intervals (CIs) for each stratum ranked from lowest to highest for fever.** Panel A: An intercept model with only intersectional strata; Panel B: A full model after adding main effects (variables used to create intersectional strata) with intercept model. Negative values indicate lower risk (privileged groups) of fever in under-five 'than expected' based on the additive contributions of the main effects, while positive values indicate higher risk (disadvantaged groups) of fever in under-five 'than expected' for that stratum.

for Model 1 ranging from -1.12 to 1.09 with only 2 (0.45%) of 441 intersectional strata's 95% CI not crossing zero (on the far left and far right), indicating significant intersectional effects estimates. Similarly, the Panel B represents intersectional effects for Model 2 ranging from -0.58 to 0.65, with all their 95% CI having crossed zero. This indicates that although intersectional effects are present for fever in U5y, reduced after controlling for main effects.

**ARI.** For Model 1, the VPC was 4.2% indicating a low DA. That is only 4.2% of the total individual variance among U5y with and without ARI was attributable to the intersectional strata level (Table 1). The AUC-ROC was 72.9%, which is an acceptable DA implying that intersectional strata have good explanatory power to discriminate U5y with and without ARI.

The VPC and the AUC-ROC dropped to 1.3% and 66.4%, respectively, after adjusting for main effects variables (e.g., child age, division, wealth index, age of household head, types of cooking fuel used, and mother's employment) in Model 2. The non-zero VPC indicates the existence of moderate multiplicative effects of intersectional strata based on the main effects variables included in the model. The reduced AUC-ROC (66.4%) suggests a low DA of intersectional strata to discriminate U5y with an ARI from those without an ARI. The PCV was 69.8%, indicating that 30.2% of the total individual variance was not explained by the main effects, which can be attributable to intersectional effects in Model 2.

Division and age of the household head were significantly associated with ARI in U5y (Table 1). Children living in slums in Khulna and other divisions were at higher risk of getting ARI compared to children living in Dhaka slums. Children from households with older household heads (aged 30 to 44) had 34% lower risk of having ARI compared to children from households with younger household heads (aged 13 to 29) (Table 1).

Fig 4 presents estimates of intersectional effects (stratum-level residuals) for ARI with their corresponding 95% CI ranked from lowest to highest. The Panel A shows intersectional effects

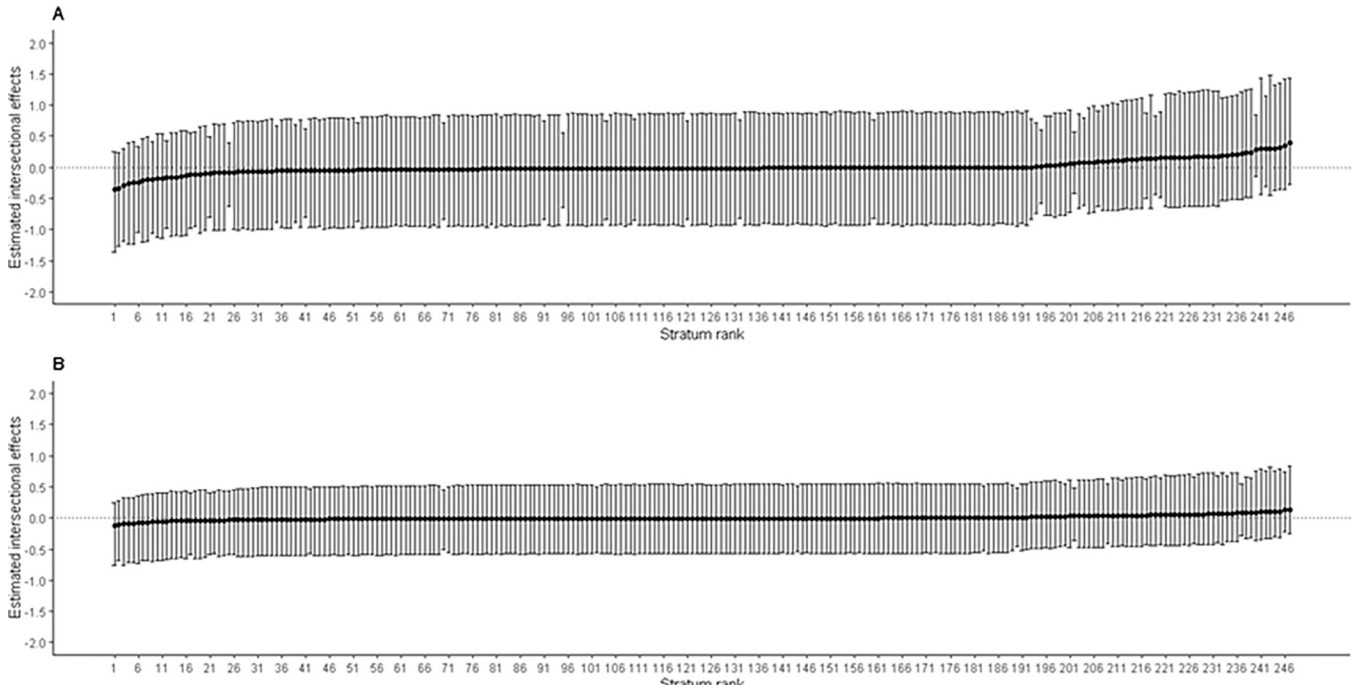

**Fig 4. Estimated intersectional effects (stratum-level residuals) and their corresponding 95% credible intervals (CIs) for each stratum ranked from lowest to highest for acute respiratory infections (ARI).** Panel A: An intercept model with only intersectional strata; Panel B: A full model after adding main effects (variables used to create intersectional strata) with intercept model. Negative values indicate lower risk (privileged groups) of ARI in under-five 'than expected' based on the additive contributions of the main effects, while positive values indicate higher risk (disadvantaged groups) of ARI in under-five 'than expected' for that stratum.

estimates for Model 1 ranging from -1.36 to 1.48, with all their 95% CI crossing zero (on far left and far right), indicating non-significant intersectional effects estimates. Similarly, the Panel B represents intersectional effects for Model 2 ranging from -0.76 to 0.84, with all their 95% CI having crossed zero, suggesting the intersectional estimates are not significantly different across intersectional strata (Fig 4).

## Discussion

While previous studies mostly investigated the association between social determinants and health outcomes, this study has extended those analyses by evaluating the association of collective SDoH and health outcomes for children under five years old (U5y) via an intersectional approach. Using MAIHDA approach we find evidence of moderate intersectional effects estimates on children's health condition based on social attributes. The PCV values (e.g., 85.90%, 78.42% and 69.77% for cough, fever and ARI, respectively) suggest that some differences in U5y health outcomes can be attributed to intersectional effects [51]. Based on the analyses, we have found that some children in certain intersectional strata are more vulnerable compared to others and therefore need a greater level of interventions compared to other groups despite all coming from an already marginalized population of urban slum dwellers.

For example, older children (e.g., 2 to 5 years) with younger household heads (e.g., 13 to 29 years), households having better waste disposal system (bin outside) and unemployed mothers and U5y living in Dhaka division are more vulnerable to having a cough. Similarly, older children (e.g., 2 to 5 years) with older household heads (e.g., 30 to 44 years) and from households having better waste management system (collected from home) and separate kitchens, employed mothers aged 18 years and above living in Dhaka Division are more vulnerable to having a fever. Older children (e.g., 2 to 5 years) from households of middle wealth quintile with older household heads (30 to 44 years) and unemployed mothers, households using natural gas and U5y living in Dhaka Division are more vulnerable to having an ARI.

The combined divisions of Barisal, Chittagong, Rangpur and Sylhet had higher risk of cough, fever and ARI than those living in Dhaka Division. Similarly, U5y living in Rajshahi and Khulna divisions were more likely to have cough and ARI, respectively, compared to those living in slums in in Dhaka Division (Fig 5). These findings are consistent with previous studies [8, 27].

This regional variation in risks might be due to temperature, indoor air pollution, access to healthcare facilities, and fuels used for cooking [25]. According to Bangladesh Child Well-Being Survey 2016, the usage of solid fuels (wood and charcoal) for cooking was found to be the lowest in Dhaka Division (29.1%) compared to Barisal (82.9%), Chittagong (45.2%), Khulna (79.1%), Rajshahi (78.9%), Rangpur (88.9%) and Sylhet (47.7%) [6]. Combustion of solid fuels for cooking releases health damaging pollutants and children are the most affected as exposure to biomass smoke is associated with ARI [52]. Unlike other divisions, non-government organisations and informal private-for-profit providers have been dominating in Dhaka slums [53]. Additionally, slum dwellers in Dhaka have relatively stable sources of income (e.g., readymade garments and service sectors) and higher income compared to those in Rajshahi (e.g., day laborers—both agricultural and non-agricultural laborers, domestic maids etc.). Additionally, children in financially well-off families are more likely to have greater access to healthcare services both preventive and promotive compared to those from impoverished families [54]. These factors might partly explain the higher risk of childhood illnesses among U5y living in Rajshahi, Khulna and other divisions compared to those living in Dhaka Division.

We have also found that being an older child (2 to 5 years) appeared to be a protective factor against cough which is similar to previous studies in the country [8, 14]. Rahman and Hossain

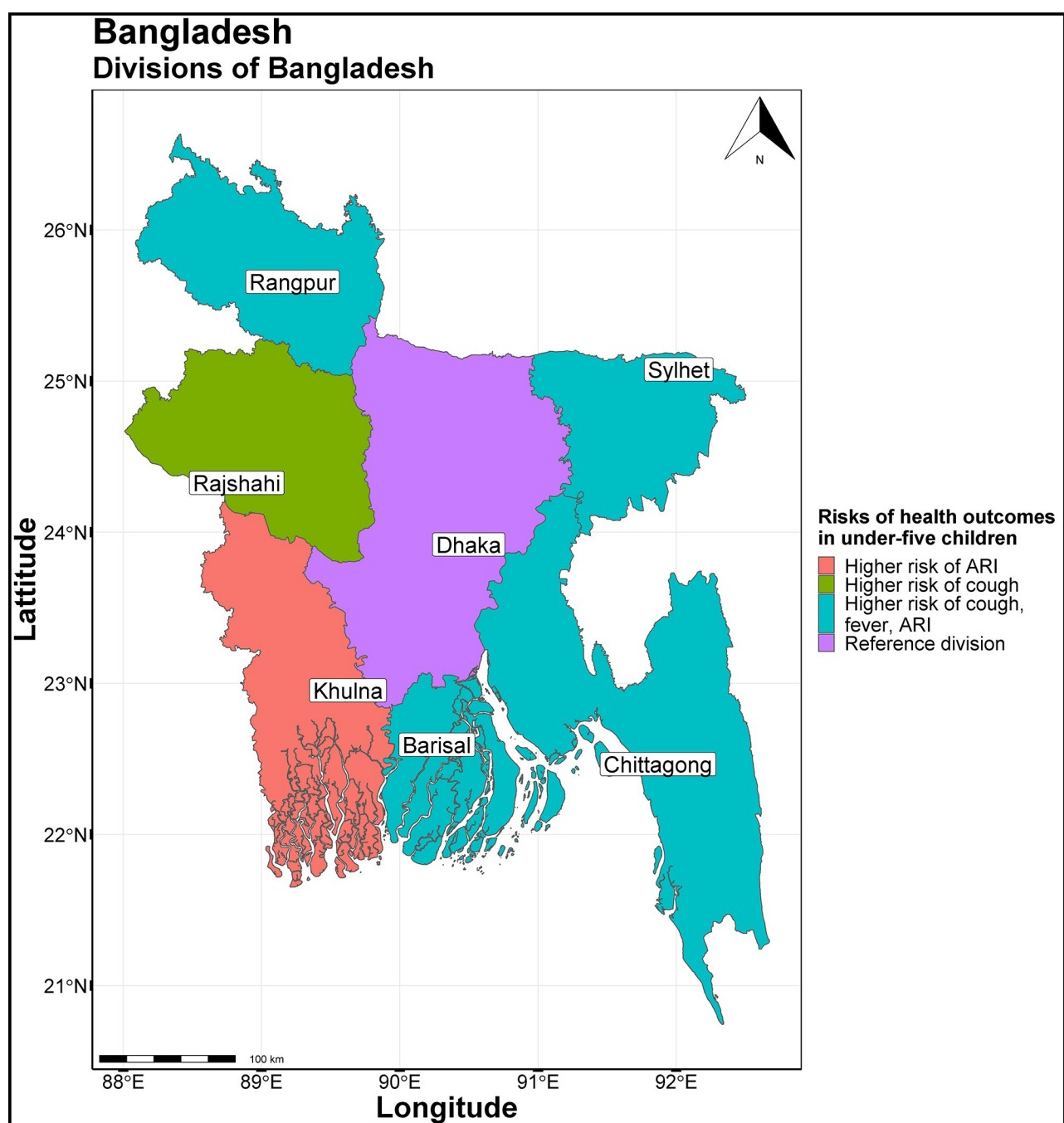

**Fig 5. Health outcomes of children under five years old (U5y) living in slums in different divisions of Bangladesh.** R software was used to create the map. The source of the basemap shapefile is available on GADM website at: https://geodata.ucdavis.edu/gadm/gadm4.1/shp/gadm41_BGD_shp.zip. The data are freely available for academic use and other non-commercial use (CC-BY 2.0). The GADM license is available at https://gadm.org/license.html.

reported that older children have a lower risk of developing diarrhoea, fever and ARI [8], while Kundu et al. found that younger children were more likely to develop both diarrhoea and ARI compared to their older counterparts [14]. The possible reason is that as children get

older, they develop immunity and know how to interact better with their caregivers and eat healthy foods, leading to reducing chances of incidence of childhood [55–57].

Our findings show that maternal employment is associated with better child health outcomes such as children are less likely to suffer if their mothers are employed which is consistent with previous studies [58, 59]. We have also found those households which are headed by older persons to have lower prevalence of cough, fever, and ARI in children, which is consistent with a previous study [60]. Finally, our study also shows that children from households with better garbage disposal systems present improved child health outcomes (e.g., cough and fever) which is consistent with a previous study that reported children living in households with better garbage disposal systems had lower prevalence of respiratory infections such as cough or nasal discharge in the past 6 months [15]. Households having better garbage disposal systems may indicate relatively well-off households with better knowledge of parenting and child development and environmental awareness who might take better care of their children leading to better child health outcomes [61].

## Strengths and limitations

Our study has several strengths as follows. First, to the best of our knowledge, this is the first study to apply quantitative intersectionality to understand SDoH that drivers of health outcomes for U5y living in slums in Bangladesh. The use of quantitative intersectionality informed evidence will bridge the knowledge gap and inform policy. Second, while measures of discriminatory accuracy such as VPC, PCV, and AUC-ROC are extensively used in clinical epidemiology to identify new biomarkers for diseases are rarely used in public health and social epidemiology [23, 62–64]. The application of these measures in our study to investigate how social positions drive health outcomes for U5y in Bangladesh slums will lead better interventions among these marginalised population. Third, the most salient advantage of the MAIHDA method is its improved and inductive analysis with numerous stratum-specific interactions or stratum-level residuals in a model that includes the main effects of the variables that define the stratum [23, 48]. For instance, we created 441 strata for fever in combination of eight main effect variables and the number of observations within an intersectional stratum have no impact on the MAIHDA model as stratum with few observations can pool information from others via multilevel modelling to obtain better inferences [38].

However, our study has some limitations. First, the UHS 2013 is nine years old. Therefore, our results may need to be interpreted with caution as the socioeconomic position of the slum population may have changed over time due to the increasing size and density of slums [65]. Second, our analyses are based on only those cases in the data set without any missing values on any of the three health outcomes. We assumed that those missing values were completely at random and, therefore, not susceptible to selection bias [66]. Third, MAIDHA approach assumes no omitted variable bias after adjusting for main effects variables in the full model to measure intersectional effects [35]. However, we have selected variables based on statistical significance to keep the analysis tractable [35]. In addition, we have used secondary data that were collected for different purposes, and we had limited options to select variables. Finally, we cannot generalise our findings beyond the slums in Bangladesh.

## Conclusion and recommendations

Applying MAIHDA approach we have found that SDoH collectively affects the health outcomes of U5y moderately. In addition, we found that the slum location, mother's employment, age of household head, and the household's garbage disposal system are associated with U5y health outcomes.

MAIHDA approach helps make more comprehensively informed decisions about suitable public health action to address an issue under study. We find the low discriminatory accuracy of the intersectional strata suggesting universal interventions rather than interventions exclusively focused on strata with a higher average risk of ARI, cough or fever. From the policymaking perspective, our findings have important policymaking implications. First, our findings suggest that with moderate intersectional effects, policy makers are supposed to consider those social groups which are more disadvantaged when making policies aimed at improving U5y health outcomes. Second, policymakers should design interventions and promote protective factors in relatively high-risk areas such as slums in Rajshahi, and Khulna where children are at higher risk of having childhood illness such as cough and ARI compared to those living in Dhaka slums. As a protective factor, our findings show that maternal employment, economically productive age of household heads and better garbage management systems are associated with improved child health. Therefore, the local government institutions of the cities and towns in Bangladesh (e.g., City Corporation Mayor, Municipality or *Pourasabha* Chairman) who have responsibility and legal authority should consider these protective factors when making decisions and designing interventions for improving U5y health. Finally, while U5y living in urban slums with precarious and crowded environments frequently suffer from childhood illness, our study shows that better waste management protects children from having cough and/or fever. This finding underlines the significance of the improvement of housing quality in slum areas affected by poor waste management. Therefore, integrated efforts from government and non-government are needed for designing public health intervention to minimise the risk of diseases associated with garbage disposal systems as well as to create public awareness about hygiene practice among slum dwellers in the country.

## Supporting information

**S1 File. Statistical details.**
(DOCX)

**S1 Table. Description of health outcomes and predictor variables for the analysis for Bangladesh Urban Health Survey 2013 (UHS 2013).**
(DOCX)

**S2 Table. Distribution of socio determinants characteristics for cough.**
(DOCX)

**S3 Table. Distribution of socio determinants characteristics for fever.**
(DOCX)

**S4 Table. Distribution of socio determinants characteristics for acute respiratory infections (ARI).**
(DOCX)

**S5 Table. Univariate analyses for cough.**
(DOCX)

**S6 Table. Univariate analyses for fever.**
(DOCX)

**S7 Table. Univariate analyses for acute respiratory infections (ARI).**
(DOCX)

**S8 Table. Calculations of intersectional strata for each of three outcomes.**
(DOCX)

## Author Contributions

**Conceptualization:** Eliud Kibuchi, Alastair H. Leyland, Linsay Gray.

**Data curation:** Bachera Aktar, Sabrina Fatema Chowdhury, Imran Hossain Mithu, Zahidul Quayyum.

**Formal analysis:** Proloy Barua.

**Funding acquisition:** Eliud Kibuchi, Bachera Aktar, Sabina Faiz Rashid, Linsay Gray.

**Investigation:** Proloy Barua.

**Methodology:** Proloy Barua, Eliud Kibuchi, Zahidul Quayyum.

**Project administration:** Proloy Barua, Bachera Aktar, Sabrina Fatema Chowdhury, Imran Hossain Mithu.

**Software:** Proloy Barua.

**Supervision:** Eliud Kibuchi.

**Validation:** Eliud Kibuchi, Bachera Aktar, Zahidul Quayyum.

**Visualization:** Proloy Barua.

**Writing – original draft:** Proloy Barua.

**Writing – review & editing:** Proloy Barua, Eliud Kibuchi, Bachera Aktar, Sabrina Fatema Chowdhury, Imran Hossain Mithu, Zahidul Quayyum, Noemia Teixeira de Siqueira Filha, Alastair H. Leyland, Sabina Faiz Rashid, Linsay Gray.

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
