## [Decision Letter · Decision Letter 0]

22 Nov 2022

PGPH-D-22-01685

The effects of social determinants on children’s health outcomes in Bangladesh slums through an intersectionality lens: an application of multilevel analysis of individual heterogeneity and discriminatory accuracy (MAIHDA)

Dear Dr. Barua,

Thank you for submitting your manuscript to PLOS Global Public Health. After careful consideration, we feel that it has merit but does not fully meet PLOS Global Public Health’s publication criteria as it currently stands. Therefore, we invite you to submit a revised version of the manuscript that addresses the points raised during the review process.

Please review the manuscript closely and address all the areas as outlined below. Please focus your efforts on the statistical analysis, definitions (especially intersectional) and ensure that you are using unbiased language throughout.

We look forward to receiving your revised manuscript.

Kind regards,

Ashti Doobay-Persaud

Academic Editor

Journal Requirements:

1. Please amend your detailed online Financial Disclosure statement. This is published with the article. It must therefore be completed in full sentences and contain the exact wording you wish to be published.

Please state what role the funders took in the study. If the funders had no role in your study, please state: “The funders had no role in study design, data collection and analysis, decision to publish, or preparation of the manuscript.”

Reviewers' comments:

Reviewer's Responses to Questions

**Comments to the Author**

1. Does this manuscript meet PLOS Global Public Health’s publication criteria? Is the manuscript technically sound, and do the data support the conclusions? The manuscript must describe methodologically and ethically rigorous research with conclusions that are appropriately drawn based on the data presented.

Reviewer #1: Yes

Reviewer #2: Partly

2. Has the statistical analysis been performed appropriately and rigorously?

Reviewer #1: I don't know

Reviewer #2: No

3. Have the authors made all data underlying the findings in their manuscript fully available (please refer to the Data Availability Statement at the start of the manuscript PDF file)?

Reviewer #1: Yes

Reviewer #2: No

4. Is the manuscript presented in an intelligible fashion and written in standard English?

Reviewer #1: Yes

Reviewer #2: Yes

5. Review Comments to the Author

Reviewer #1: Overall, the authors have worked diligently to conduct a statistical analysis using MAIHDA to evaluate the effects of SDoH on child health outcomes in Bangladeshi slums. I was not familiar with this statistical methodology prior to reviewing this work, which may contribute to my relative lack of understanding of its application here. While I believe this work could be impactful, this manuscript needs revision in a few aspects before it is ready for publication.

1. It is unclear why cough is separate from ARI as a cause of mortality. Can the authors say more about how they chose these three health outcomes to evaluate? Why was diarrhea not included? The paragraph in the introduction where the causes of child mortality is introduced is a bit hard to follow, as it begins by stating that fever, cough, and ARI are common causes of child mortality but then goes on to discuss prevalence of ARI and then return to mortality in the middle of the paragraph, then back to ARI hospitalization rate. Would rework the introduction to be more direct and clearly state why these three health outcomes were selected.

2. It is not clear to me how the intersectional strata are defined. It is entirely possible that this is due to my personal unfamiliarity with this methodology, but the methods section does not clearly explain this to the average researcher.

3. The statistical analysis reported varied factors for each of the three health outcomes. It is unclear why certain social determinant factors were reported for cough vs fever vs ARI – it makes it difficult to understand the impact of SDoH on health outcomes in aggregate if it not clearly stated which determinant factors are included in each analysis.

4. The conclusions drawn in the discussion are not necessarily supported by this evidence as there is moderate and disparate effects of SDoH that can be drawn for each of the three health outcomes. Also, it is unclear how these analyses provide additional meaningful information beyond what has been published about the association of SDoH and health outcomes in pediatric populations, as the authors reference extensively in the introduction.

5. The authors reference risks of each health outcome relative to each division of slums. Perhaps it would be more illustrative to use a map to highlight the significant risks present in each division, to allow the reader to create a relational understanding of the risks in Dhaka slums vs others around the country (which is referenced in the conclusions and policy implications).

6. Try to avoid language that could be viewed as insulting or pejorative. For example, would not use the phrase “slum people”.

7. At times, the word choice and grammar are confusing and unclear, which contributed to the difficulty I had in understanding the text. I would recommend reviewing for grammar, typos, and clarity.

Reviewer #2: The study aims to examine the effects of social determinants of children’s health outcomes in slums in Bangladesh. Association between individual social determinants and health outcomes have previously been analyzed and established; arguably, the authors method adds value by looking at the same question from an intersectional approach. I enjoyed reading the piece, and would like to highlight a few other areas where the manuscript can be strengthened with regard to its scientific merit.

1) Avoid causal language throughout the manuscript. There shouldn’t really be any reference to “effects” since this is an associational study.

2) There needs to be a much clearer discussion—in the introduction—of what the authors mean by intersectional. The authors never fully explain it anywhere.

3) Related to (2), please explain the advantage of using MAIHDA compared to using a general approach (regression with interaction terms for various combinations of the independent variables). The distinction between the two—and the justification for using MAIHDA—is important because the authors have categorized several variables that could have been used as continuous variables—thus retaining and utilizing the variation in the variable—if they had not used MAIHDA. Also, my understanding is that it is relatively straightforward to obtain marginal effect of interaction terms in regression models (essentially getting at the intersectional aspect, from what I understand), so I am not clear what MAIHDA buys us here.

4) Also related, instead of picking variables for strata based on statistical significance in univariate analysis, why not base it on existing literature and on authors’ understanding of the context? Or, based on variables over which policymakers have some control (e.g., employment, education, housing conditions)?

5) On page 8, the authors need to explain the method more clearly. S1 File is useful. I wish the part in the main text was written in more plain language.

6) In the methods (and in the method appendix), there is reference to “assuming no relevant variables are omitted suggesting presence of interaction effect”. This seems like a big assumption. How do we know that it holds?

7) Stylistically, please consider reorganizing the results section. Results in 3 outcomes are explained separately. Tables 1, 2 and 3 can be combined into one table, and the results can be explained more coherently—focusing on the key points.

8) Minor: based on the findings of the study, what do we now know that we did not from so many previous studies that have been conducted on this topic (at least from the policy perspective in Bangladesh)? Some recommendations are made in the Conclusion, but they seem too general to be useful.

Other minor points:

• Line 132 – the authors likely mean the quintiles of wealth index (of from S1 – categories of wealth index), not wealth index, as the method requires categorical variables

• Line 158-159 – incomplete sentence

• Line 164 – “1” missing

• Line 165 – Is it 0.5 to 1 or 0.05 to 1? The previous sentence says 0.5.

• Line 186 – I’d put the link to the 2013 Urban Health Survey page, not the general link to UNC’s dataverse.

• Line 197 – Explain how you created 147 strata for cough. There seem to be 9 coefficients that are statistically significant. How do we go from those 9 coefficients to 147 strata? Similar explanation can work for how 441 and 247 strata were created for fever and ARI.

• Table 1 – Significant at what level? Also, the authors likely mean *statistically* significant.

• Table 1 – Strata variance in what? Is this for outcome?

• In S3/S4 Table, mothers with no education seem to be in both ‘Mother Ever Attended School’ and ‘Mother’s Highest Education’. The first one seems redundant.

6. PLOS authors have the option to publish the peer review history of their article (what does this mean?). If published, this will include your full peer review and any attached files.

**Do you want your identity to be public for this peer review?** For information about this choice, including consent withdrawal, please see our Privacy Policy.

Reviewer #1: No

Reviewer #2: No

---

## [Decision Letter · Decision Letter 1]

13 Jan 2023

PGPH-D-22-01685R1

The effects of social determinants on children’s health outcomes in Bangladesh slums through an intersectionality lens: an application of multilevel analysis of individual heterogeneity and discriminatory accuracy (MAIHDA)

Dear Dr. Barua,

Thank you for submitting your manuscript to PLOS Global Public Health. After careful consideration, we feel that you have made many changes that are acceptable however some minor revisions still need to be made to meet publication criteria for PLOS global health. Therefore, we invite you to submit a revised version of the manuscript that addresses the points raised during the review process.

We look forward to receiving your revised manuscript.

Kind regards,

Ashti Doobay-Persaud

Academic Editor

Journal Requirements:

2. Please insert an Ethics Statement at the beginning of your Methods section, under a subheading 'Ethics Statement'. It must include:

1) The name(s) of the Institutional Review Board(s) or Ethics Committee(s)

2) The approval number(s), or a statement that approval was granted by the named board(s) 

3) (for human participants/donors) - A statement that formal consent was obtained (must state whether verbal/written) OR the reason consent was not obtained (e.g. anonymity). NOTE: If child participants, the statement must declare that formal consent was obtained from the parent/guardian.

3. Fig 4: please (a) provide a direct link to the base layer of the map (i.e., the country or region border shape) and ensure this is also included in the figure legend; and (b) provide a link to the terms of use / license information for the base layer image or shapefile. We cannot publish proprietary or copyrighted maps (e.g. Google Maps, Mapquest) and the terms of use for your map base layer must be compatible with our CC-BY 4.0 license. 

Additional Editor Comments (if provided):

Reviewers' comments:

Reviewer's Responses to Questions

**Comments to the Author**

1. If the authors have adequately addressed your comments raised in a previous round of review and you feel that this manuscript is now acceptable for publication, you may indicate that here to bypass the “Comments to the Author” section, enter your conflict of interest statement in the “Confidential to Editor” section, and submit your "Accept" recommendation.

Reviewer #1: (No Response)

Reviewer #2: All comments have been addressed

2. Does this manuscript meet PLOS Global Public Health’s publication criteria? Is the manuscript technically sound, and do the data support the conclusions? The manuscript must describe methodologically and ethically rigorous research with conclusions that are appropriately drawn based on the data presented.

Reviewer #1: Yes

Reviewer #2: Yes

3. Has the statistical analysis been performed appropriately and rigorously?

Reviewer #1: Yes

Reviewer #2: Yes

4. Have the authors made all data underlying the findings in their manuscript fully available (please refer to the Data Availability Statement at the start of the manuscript PDF file)?

Reviewer #1: Yes

Reviewer #2: Yes

5. Is the manuscript presented in an intelligible fashion and written in standard English?

Reviewer #1: No

Reviewer #2: Yes

6. Review Comments to the Author

Reviewer #1: 1. Thank you for the edits and thoughtful responses to my initial comments. I believe you have addressed the majority of my questions; however, the text still requires copyediting for grammar. A few instances from the introduction are listed below, though this is not comprehensive as copyediting is not the specific role of a reviewer. Some common errors noted are missing articles (for example, "the"), plural vs singular conjugation, and verb-tense agreement. Would recommend a *detailed* review specifically for grammar.

Line 59: "In Bangladesh context, most widely accepted definition..." should read "In the Bangladeshi context, the most widely accepted definition..."

Line 63: should read, "according to the Census of..."

Line 69: should read, "there is also regional variation in [prevalence? incidence?] of childhood illness..."

Line 74: should read, "among males compared to females"

Line 81: "in future" should be "in the future"

Line 96: should read "they lead to aggravated social inequities"

2. I appreciate the addition of the map. There is a typo in the "reference division" label that should be corrected.

Reviewer #2: Thank you for addressing the substantive comments. I do have reservations about picking variables based on statistical significance but I understand if that is consistent with practice in this line of the literature. "It was not possible to include all variables because our analyses included main effects that only showed statistical significance (a p-value less than 0.05) on univariate analysis" (page 9 of the manuscript) is circular because that was a decision you made in the first place. Perhaps a better justification would be that you wanted to keep the analysis tractable.

7. PLOS authors have the option to publish the peer review history of their article (what does this mean?). If published, this will include your full peer review and any attached files.

**Do you want your identity to be public for this peer review?** For information about this choice, including consent withdrawal, please see our Privacy Policy.

Reviewer #1: No

Reviewer #2: No

---

## [Editor Report · Decision Letter 2]

3 Feb 2023

The effects of social determinants on children’s health outcomes in Bangladesh slums through an intersectionality lens: an application of multilevel analysis of individual heterogeneity and discriminatory accuracy (MAIHDA)

PGPH-D-22-01685R2

Dear Dr. Barua,

We are pleased to inform you that your manuscript 'The effects of social determinants on children’s health outcomes in Bangladesh slums through an intersectionality lens: an application of multilevel analysis of individual heterogeneity and discriminatory accuracy (MAIHDA)' has been provisionally accepted for publication in PLOS Global Public Health.

Thank you again for supporting Open Access publishing; we are looking forward to publishing your work in PLOS Global Public Health and for your patience.

Best regards,

Ashti Doobay-Persaud

Academic Editor